# Application of Machine Learning for Automating Behavioral Tracking of Captive Bornean Orangutans (*Pongo Pygmaeus*)

**DOI:** 10.3390/ani14121729

**Published:** 2024-06-08

**Authors:** Frej Gammelgård, Jonas Nielsen, Emilia J. Nielsen, Malthe G. Hansen, Aage K. Olsen Alstrup, Juan O. Perea-García, Trine H. Jensen, Cino Pertoldi

**Affiliations:** 1Department of Chemistry and Bioscience, Aalborg University, Frederik Bajers Vej 7H, 9220 Aalborg, Denmark; jnielc21@student.aau.dk (J.N.); mgha20@student.aau.dk (M.G.H.); ejni20@student.aau.dk (E.J.N.); trine@bio.aau.dk (T.H.J.); cp@bio.aau.dk (C.P.); 2Department of Nuclear Medicine & PET, Aarhus University Hospital and Department of Clinical Medicine, Aarhus University, Palle Juul Jensens Boulevard 99, 8000 Aarhus, Denmark; aagealst@rm.dk; 3Faculty of Social and Behavioural Sciences, Leiden University, 2333 Leiden, The Netherlands; juan.olvido@gmail.com; 4Aalborg Zoo, Mølleparkvej 63, 9000 Aalborg, Denmark

**Keywords:** orangutans, object detection, image classification, zoological institution, artificial intelligence, automatic behavior tracking

## Abstract

**Simple Summary:**

This study investigates the application of machine learning in the form of image classification and object detection to video material to automate behavior recognition in captive orangutans (*Pongo pygmaeus*). A machine learning model was constructed using a 2 min video consisting of 30 s clips of each selected behavior. The machine learning model had a 13% detection rate and showed potential for future expansion, with the goal of automating behavioral studies, but also notable issues that should be considered when using such methods.

**Abstract:**

This article applies object detection to CCTV video material to investigate the potential of using machine learning to automate behavior tracking. This study includes video tapings of two captive Bornean orangutans and their behavior. From a 2 min training video containing the selected behaviors, 334 images were extracted and labeled using Rectlabel. The labeled training material was used to construct an object detection model using Create ML. The use of object detection was shown to have potential for automating tracking, especially of locomotion, whilst filtering out false positives. Potential improvements regarding this tool are addressed, and future implementation should take these into consideration. These improvements include using adequately diverse training material and limiting iterations to avoid overfitting the model.

## 1. Introduction

The orangutan (*Pongo pygmaeus*) is partly endemic to Borneo. The Bornean forests have experienced great reductions because of fires, deforestation, logging, and the establishment of plantations. Due to illegal hunting and a reduction in suitable habitats, the species is now listed as critically endangered, protected by law, and part of conservation programs all over the world [1]. Conservation programs are in part facilitated by zoological associations such as the Danish, European and World Associations of Zoos and Aquaria (DAZA, EAZA and WAZA) that uphold requirements of their members such as prioritizing research to continuously uphold good welfare standards [2,3,4].

To properly understand the implications of orangutan behavior and welfare, it must first be understood what constitutes healthy behavior. This is not easily achieved in captive animals, since the natural behavior of wild orangutans does not inherently constitute healthy behavior for captive ones. This is both because not all wild behaviors are desirable in a captive setting, and because not all wild behaviors are reproducible in captivity [5]. Stereotyped behaviors in captive individuals may be assessed and analyzed in a behavioral study as indicators of welfare [6]. Continuous focal sampling is useful for analyzing behavior and understanding which behaviors are healthy, such as spending a lot of time foraging or interacting with enrichment, and which behaviors are considered pathological, such as stereotypies [7,8].

In recent years, artificial intelligence (AI) has been a useful tool for improving video and image analysis, through applying machine learning to automatically identify and differentiate between species, individuals, and/or behaviors [9,10,11,12]. The relevant parts of machine learning for video and image analysis are image classification and object detection [9,13,14]. Image classification may identify which orangutan is shown in a frame, based on a trained model [10,14]. Object detection may be used for identifying and labeling certain behaviors executed by an orangutan whilst giving information about the spatial coordinates and time (Figure 1) [10,15].

Knowledge of the coordinates as a function of time may allow for further analysis, such as determining whether the subject is lying down, standing up, moving, etc. Creating a model capable of tracking certain behaviors would thus make behavioral analysis more effective, by allowing for automatization and thus reducing the work hours required for behavioral coding [9,16]. Previous studies have also demonstrated the capabilities of using machine learning models for behavior tracking in wild and captive primates, with practical applications such as automated audiovisual action recognition, recognizing minute behavior differences in still images of primates, and allowing for behavior observation on a 24 h basis [12,17,18,19]. Congdon et al. (2022) studied the possibility of detecting orangutans in image frames and could distinguish the subjects with relatively high accuracy, although the following stage of this process, consisting of recognizing orangutan behavior, is still in progress [12]. Bain et al. (2021) present the novel use of audio and visual recordings to determine the audiovisual behaviors ‘nut cracking’ and ‘drumming’ among wild chimpanzees, although this study relies on audio as a prescreening method to find desired audiovisual behavior and does not aim to cover a broader spectrum of behaviors [17]. Feng et al. (2023) successfully utilized deep learning to distinguish a broader spectrum of behaviors in still images of primates, but only used this model on testing datasets gathered from public image repositories containing only those selected behaviors [18]. These studies have not been used for continuous focal sampling in captive primates using CCTV footage, which may eventually allow for somewhat replacing manual continuous focal sampling for behavioral analysis, although this may be studied further in the near future [11,12]. Developing such a model may also aid in providing a method that is consistent and standardized across studies and is less error-prone than manual coding by researchers [11,19].

The aim of this study was to investigate the applicability of computer vision by automating the tracking of certain behaviors and recognizing individuals from CCTV video material, to evaluate the potential for greater efficiency in this type of research in the future.

## 2. Materials and Methods

### 2.1. Subjects and Enclosure

The behavior of two adult Bornean orangutans at Aalborg Zoo (Denmark) was examined. The male orangutan was born in the Zoo Aquarium de Madrid, Spain, in 2010, and the female orangutan was born at Sóstó Zoo, Hungary, in 2013. They were transferred to Aalborg Zoo in June 2022 and immediately placed in a recently renovated enclosure as the first two orangutans to inhabit the new enclosure. In August 2022, two Asian small-clawed otters (*Aonyx cinereus)* were introduced to the enclosure. The orangutans were separated in the enclosure until March 2023, after which they were introduced to each other and mostly kept together. The enclosure consists of four indoor areas approximating 577 m^3^ and two large outdoor areas approximating 2570 m^3^. Visitors could observe the orangutans through glass panels in the indoor areas and through a net surrounding the outdoor areas. The enclosure provided the orangutans with various climbing options such as ropes, swings, sway poles, artificial trees, and the surrounding net of the outdoor areas. Furthermore, the zookeepers occasionally provided the orangutans with diverse nest materials such as wood shavings and blankets. The orangutans’ diet consisted of fruit, vegetables, and an assortment of treats along with leaves and bark. Every morning, the female was given juice with a birth control pill. They were mainly fed in the indoor areas once a day. Moreover, the zookeepers sporadically fed the orangutans through the outdoor net or metal doors in the indoor and outdoor areas. Paper bags and cardboard tubes with treats were thrown onto the top net of their outdoor enclosures to stimulate the orangutans once or several times a day. Additional enrichment such as treats, honey, and oatmeal hidden in holes in trees or containers were provided regularly to increase the time spent by the orangutans on foraging. To prevent anticipatory behavior, they were not fed systematically, and to induce further variation, the zookeepers changed the configuration of open and closed gates in the orangutans’ enclosure daily.

### 2.2. Data Collection

An ethogram was constructed based on a previous behavioral study of the same orangutans [20], conversations with the zookeepers, and personal observations made at Aalborg Zoo. The ethogram was modified with the intent of only studying recognizable behaviors with an object detection model (Table 1).

The video material was collected from two periods (12 June 2023 to 16 June 2023 and 17 July 2023 to 21 July 2023) during visitor opening hours (10 a.m. to 7 p.m.). The average temperatures and precipitation for the periods were 16.4 °C with 27.0 mm rain for June and 15.9 °C with 140.8 mm rain in July [21]. The outdoor areas were monitored by two Milesight Mini PTZ dome 2.0 MP Starlight Cameras placed outside the surrounding net. An ethogram was used to manually conduct behavioral detection, where the video material was studied in GridPlayer version 0.5.3 and noted in Excel version 16.65 (22091101). This was carried out to allow for comparisons between manual observations and observations made by the automatic object detection model. A total of 12 h of video material was collected for this comparison.

Video material provided for image classification was collected on October 10th, October 17th, and October 19th with 12 MP cameras from an iPhone 11, in order to obtain high resolution images of the animals’ bodies and faces.

To demonstrate the capabilities of machine learning, an object detection model trained to recognize selected behaviors (Table 1) was constructed. The model was constructed in Rectlabel by extracting frames from a 2 min video. The 2 min video consisted of 5–10 s individual clips, so each behavior (Locomotion, Inactive, Covered Inactive and Foraging/Feeding) was displayed for 30 s each, in different places and positions. Frames from the video were extracted in Rectlabel, which gave a total output of 3331 image frames. Every tenth frame was labeled in Rectlabel with a specific behavior, as seen in Figure 1. The resulting 334 labeled images were used to train a model in Create ML with 9000 iterations. The model constructed in Create ML was used to auto-label the remaining image frames. The object detection model could automatically label 3281 images frames (98.5%). All image frames left unlabeled after the autolabeling process were manually coded with the exhibited behavior. The behavior Inactive was labeled 887 times, Covered Inactive was labeled 835 times, Foraging/Feeding was labeled 814 times, and Locomotion was labeled 795 times. The new model with a total of 3331 labeled image frames was trained in Create ML with 18,000 iterations. This final model was used to analyze 12 h of video material.

### 2.3. Data Analysis

The collected data from the manual observations and automated observations were analyzed using Excel version 2403 (Build 16.0.17425.20124). The machine learning computations were conducted in RectLabel version 2023.10.22 (2023.10.22) and Create ML version 5.0 (121.1).

Firstly, we examined whether an image classification model could effectively distinguish between the male and female orangutan. The video material was divided into short videos of 5–10 s in QuickTime Player version 10.5 (1170.4.13.1) and subsequently converted into image frames in RectLabel, which either maintained the background or had the background removed with Preview version 11.0 (1056.2.4). The entire methodological pipeline is shown in Figure 2.

Furthermore, we examined whether an object detection model could recognize different behaviors for the female orangutan, since this subject was in view of the cameras more often. In total, 12 h of footage was analyzed with the object detection model to somewhat replicate the focal sampling used in behavioral studies. To compare these results with manual observations of the same material, time budgets were made for both methods. The methodological pipeline is shown in Figure 3.

To further analyze the data from the object detection model, the coordinates and the dimensions of the boxes were collected from the CSV file.

## 3. Results

### 3.1. Using Image Classification to Recognize Individual Orangutans

When the image classification model was trained with data pool A, consisting of 642 images (327 and 315 images of the male and female orangutan, respectively), the test accuracy was 94% when evaluated with data pool A. The image classification model trained with data pool B had a test accuracy of 95% when evaluated with data pool B. After correction, the image classification model was trained with data pool C and did not improve when evaluated with data pool B. The test accuracy, however, increased to 99% when the corrected image classification model was evaluated with data pool C. In all cases, the female was more often misclassified as the male, and further correction failed to improve the test accuracy.

### 3.2. Using Object Detection to Automate Behavioral Tracking

The object detection model (Figure 4; set to 80% confidence) labeled 13.0% of the 43483 image frames (5653 frames) in the 12 h video, which was approximately split into 1 frame per second. Of these 5653 labeled image frames, 3000 image frames (6.9% of total observation time) were labeled with ‘Locomotion’, 870 frames (2.0% of total observation time) with ‘Foraging/Feeding’, 609 frames (1.4% of total observation time) with ‘Inactive’, and 1174 frames (2.7% of total observation time) with ‘Covered Inactive’. A total of 87% of the 43,483 image frames were left unlabeled due to the orangutans being out of view (which was 41.3% of the total observation time) or because of a lack of confidence by the model.

When the 12 h video material was observed manually, 3.6% of the images were noted as ‘Locomotion’, 18.1% as ‘Foraging/Feeding’, 13.9% as ‘Inactive’, 15.6% as ‘Covered Inactive’, and 7.5% as ‘Other’. The latter category included all behaviors that were not categorized, hence why the model did not include this. The orangutans were out of view 41.3% of the time.

### 3.3. Correcting Object Detection of Locomotion

It appears from the object detection time budget (Figure 4) that Foraging/Feeding, Covered Inactive, and Inactive have much lower values than in the manual observations. It may be difficult to correct this lack of detection by the model using label box dimensions and coordinates, since there are no consistently similar tendencies to analyze. However, Locomotion has almost double the value when using the object detection model compared to that obtained using the manual ethogram, and it is possible to correct this behavioral coding using the coordinates of the label box. This higher value is found to be caused by false positives from the model. To filter out most of the false positives of Locomotion, the coordinates of the CSV file were utilized. The coordinates for each frame the model labeled as ‘Locomotion’ were plotted, as shown below (Figure 5).

The frames where movement did not occur, displayed by changes in both the x-coordinate and the y-coordinate, were labeled as false positives, instead of ‘Locomotion’. Thus, the plateaus throughout Figure 5 were removed manually. The renewed time budget including false positives for Locomotion can be seen below (Figure 6). It appears that 3.4% of the frames labeled ‘Locomotion’ by the model were false positives and 3.5% of the frames were left as ‘Locomotion’.

## 4. Discussion

### 4.1. Subject Recognition with Image Classification

The image classification model with the best test accuracy in this study had a test accuracy of 99%. However, the test accuracy decreased when the evaluation data were applied to new material, which, for practical use, would often be the case. The model should be trained with a larger data pool containing pictures of the male and female orangutans from different angles and in different lighting. The models which were trained with data pools without backgrounds (data pool B and C) more effectively distinguished the male and female orangutans, but an improved model trained with images with backgrounds would probably be preferable to implement, since this more closely resembles any practical use. Additionally, it would minimize the work effort in the training and evaluation process. Occasionally, it can be challenging for the model to identify the subject in the images, and therefore, an object detection model where the subjects are manually encapsulated by boxes could possibly be more effective at distinguishing the orangutans. Furthermore, this study was conducted on only two phenotypically different orangutans, which does not provide insight into possible limitations in distinguishing individuals of the same sex and approximate size.

### 4.2. Potential for Utilizing Label Box Dimensions to Determine Behavior

Besides using the coordinates of the label box, created by the object detection model, for correcting Locomotion, there is a variety of potential uses for the information given. An example of this is using the height/width ratio of the label box to determine simple behaviors, such as the orangutan raising its arms or lying down. This is achieved through observing frames of these behaviors and setting limits that determine when a label box’s ratio exceeds these, and thus communicates the presence of that behavior. This is exemplified below (Figure 7), where the height/width ratios of 100 frames are shown, and limits for lying down and raising arms are implemented, to visualize when these behaviors happen throughout the frames.

### 4.3. Behavior Tracking with Object Detection

The object detection model did not label 87.0% of the image frames, which indicates that the model found it difficult to recognize the four behaviors. For the behaviors ‘Foraging/Feeding’, ‘Inactive’, and ‘Covered Inactive’, the percentages varied greatly when comparing the results obtained manually to those obtained by the model. To resolve this problem, it may be favorable to lower the confidence of the model. The model was set to 80% confidence, which may have caused many of the image frames to be unlabeled due to uncertainty. As is discussed in studies by Mathis et al. (2018) and Wenkel et al. (2021), a lower confidence threshold may aid in receiving fewer false negatives at the cost of potentially increasing false positives [22,23]. For the behavior ‘Locomotion’, approximately double the percentage was obtained when using the model. This value was therefore corrected to exclude the number of false positives. Such corrections may be necessary to fully evaluate the model, and may be implemented in a multitude of ways, such as using the label box’s height and width to determine occurrence and false positives for other behaviors. These false positives might be caused by the box-labeling of the behaviors, which leads to the model mistaking a certain background used in the training as recognition of that behavior. A solution might be to label the subject with polygons rather than boxes to better encapsulate the subject and minimize the influence of the background. Another issue may be a lack of diversity in the training material, which may lead to overfitting of the model. This might also be exacerbated by using too many iterations when constructing the model. The extended object detection model showed the greatest potential for optimization when tracking ‘Locomotion’. In contrast, tracking the other behaviors of ‘Foraging/Feeding’, ‘Inactive’, and ‘Covered Inactive’ might be harder due to the lack of uniformity. Another challenge emerges in the autolabeling process when two or more different behaviors overlap, e.g., when the orangutan feeds and performs locomotion simultaneously. The model could potentially be trained with image frames with overlapping behaviors, and the behavior of interest could be labeled on these frames to overcome this challenge. An object detection model, however, can be useful for tracking certain behaviors or tendencies. Further work is necessary to assess the limitations and challenges of an object detection model in the automatization and optimization of behavioral studies.

The continued development of such models may prove a useful tool in the future for researchers and zookeepers alike. This tool may be used to quickly assess the general behavior of an animal without requiring much effort and allowing for the utilization of CCTV footage for this purpose that would otherwise already be used by zookeepers. For this reason, studying problems and possible improvements whilst constructing such models is an important and necessary first step that provides insight into the process and helps avoid similar mistakes in the future. It may also be valuable to implement other machine learning software, such as YOLO or DeepLabCut, as used by Wenkel et al. (2021) and Hardin & Schlupp (2022), respectively [23,24], to assess more specific and detailed behavior which is typical of primates.

## 5. Conclusions

This study demonstrated that the automatization of behavioral analysis with machine learning using Image Classification to recognize individual Bornean orangutans was possible and somewhat reliable. Tracking behaviors with Object Detection showed potential and was somewhat capable of labeling the behavior ‘Locomotion’. However, the Object Detection model needs further optimization.

## Figures and Tables

**Figure 1 animals-14-01729-f001:**
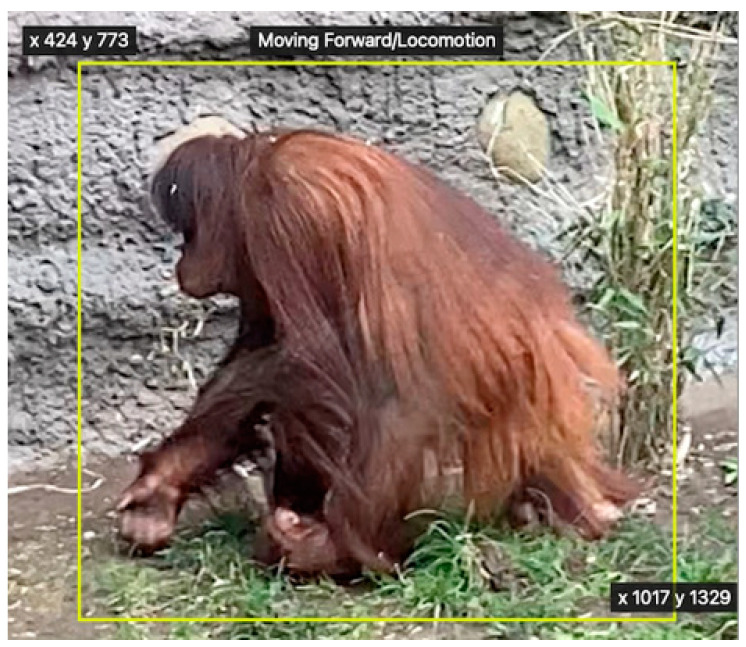
Image of the female Bornean orangutan at Aalborg Zoo, Denmark. The yellow label box specifies its location and labels the specific behavior (Moving Forward/Locomotion) with the exact coordinates (at two corners of the rectangle).

**Figure 2 animals-14-01729-f002:**
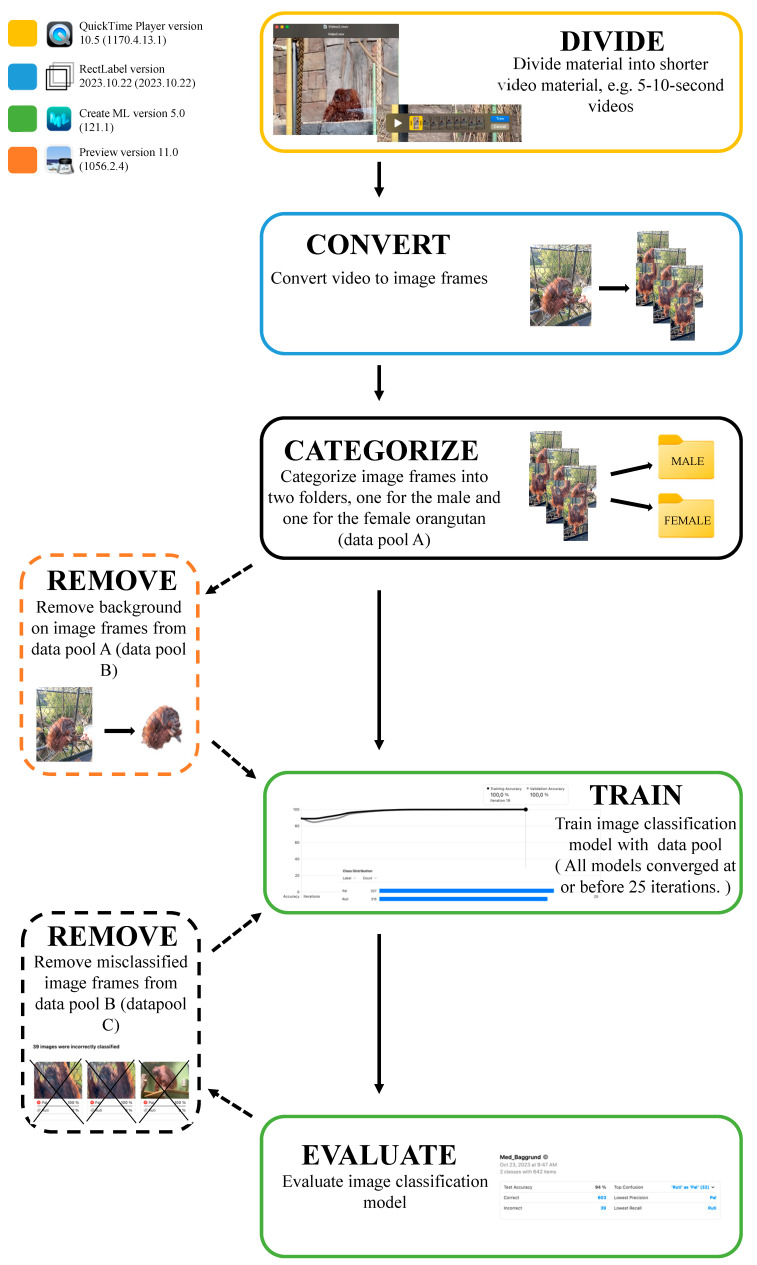
Pipeline showing the process of creating and evaluating an image classification model. The different colored boxes represent the software used.

**Figure 3 animals-14-01729-f003:**
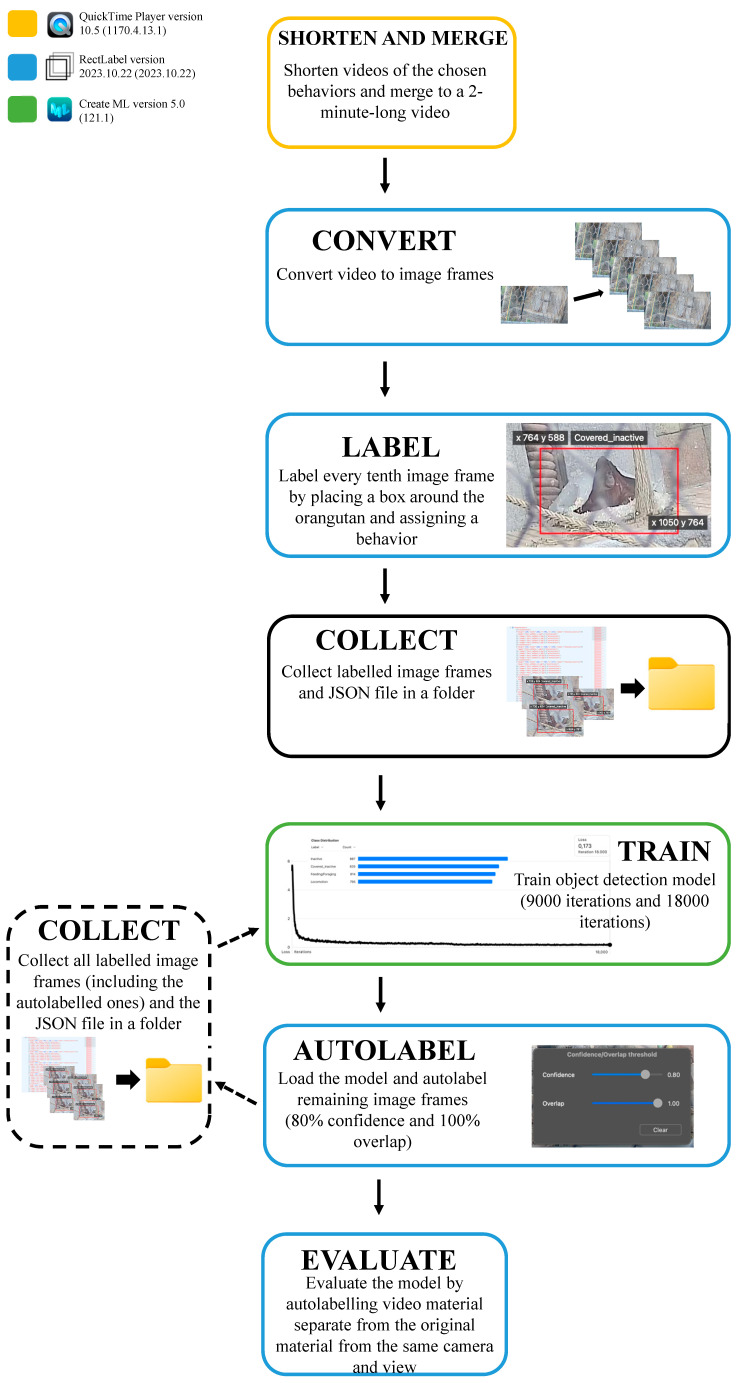
Pipeline showing the process of creating and evaluating an object detection model. The different colored boxes represent the software used.

**Figure 4 animals-14-01729-f004:**
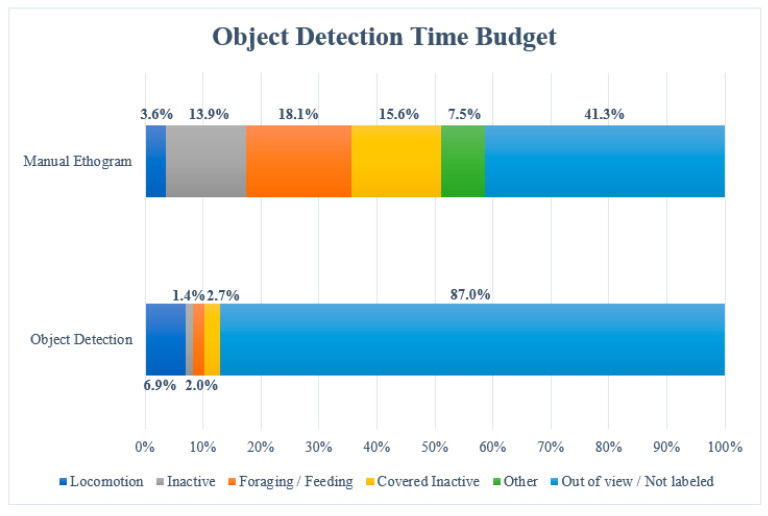
Time budget for the 12 h video when observed manually and with the object detection model. Each of the behaviors is represented by a different color, and the ethograms show percentages indicating how long the female orangutan performed each behavior.

**Figure 5 animals-14-01729-f005:**
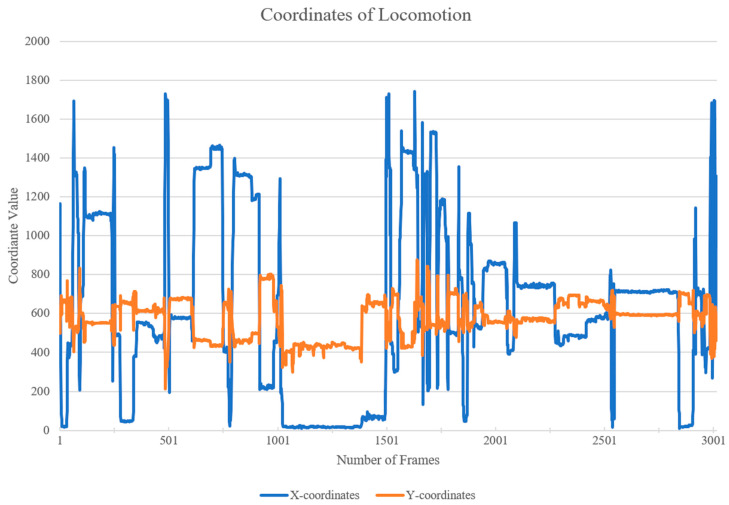
Plot of coordinates of the bottom corner of the label box. The blue line shows x-coordinates, and the orange line shows y-coordinates.

**Figure 6 animals-14-01729-f006:**
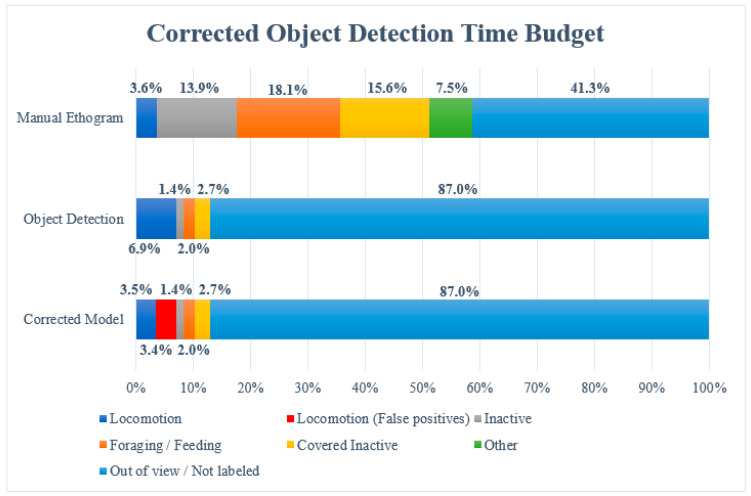
Time budget for the 12 h video when observed manually and with the object detection model, as well as data corrected for false positives in ‘Locomotion’ (shown in red). Each of the behaviors is represented by a different color, and the ethograms show percentages indicating how long the female orangutan performed each behavior.

**Figure 7 animals-14-01729-f007:**
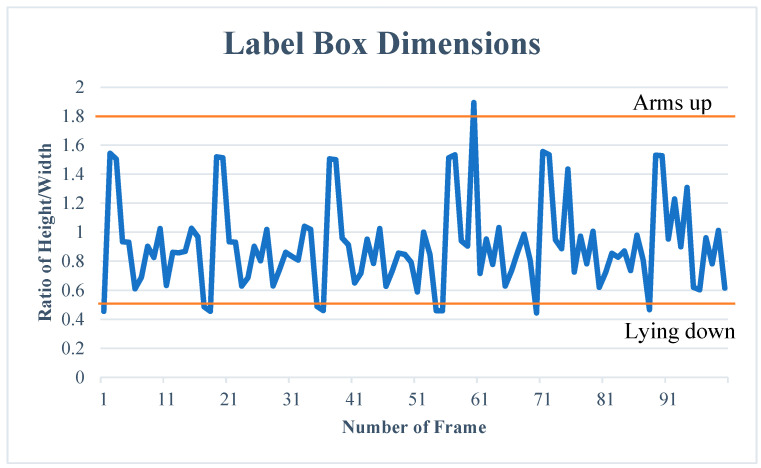
The height/width ratio of label boxes for 100 frames of footage. The upper line shows the limit of the ratio, where the behavior can be considered ‘Arms up’. The lower line shows the lower limit of the ratio, where the frame is considered ‘Lying down’.

**Table 1 animals-14-01729-t001:** Ethogram with the selected behaviors. The ethogram was modified from [20].

Behavior	Description
Locomotion	Walking, crawling, or climbing
Inactive	Stationary, e.g., sitting, lying down, or standing still
Covered Inactive	Stationary under cover, such as blankets
Foraging/Feeding	Searching for food, moving with food, or ingesting food or drinking
Out of View/Not Labeled	When it is not possible to see the orangutan because they are out of the camera’s view. Includes images that were not labeled by the object detection model
Other	Urinating, interaction with zookeepers, etc.

## Data Availability

The data presented in this study are available on request from the corresponding author.

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
