# Peer review of "Application of Machine Learning for Automating Behavioral Tracking of Captive Bornean Orangutans (Pongo Pygmaeus)"

_animals, 2024, doi:10.3390/ani14121729_

Round 1

Reviewer 1 Report (Previous Reviewer 1)

Comments and Suggestions for Authors

Much better! Now that it is clearer, there were still some things to address. The first one is the lack of information on how you will deal with potential false negatives. You have a great explanation on how to address the false positives in the locomotion detection but you fail to mention - at all - how to mitigate the false negatives. This is a huge oversight and needs to be amended. 

There were also a few things that needed to be clarified a bit because it is a lot of numbers and the reader can easily get lost within it. I noted these on the manuscript pdf. 

Finally, I still think you need to have a bit more of a "why" in your introduction. WHY is machine learning for behavior detection so useful? I know that it can "cut back on time" but there are a lot of other ways it can be useful (can potentially catch minute details humans wouldn't see, can observe for a full 24-hour period... etc.). Do some research and look at how this could practically assist the field. I also think you need a decent sized paragraph discussing what has already been done in machine learning for behavior tracking. While this is a new field, it's advancing quickly. I think you need a bit more oomph in your introduction to provide more background on machine learning in zoos. What projects have been done? What were their limitations? Did you have the same issues with your model as other projects? Did you have successes where they failed? All of these ideas should be touched on (and this is just off the top of my head - I'm sure there are other questions you can address!). 

Comments on the Quality of English Language

English was much better. There were a few spelling errors that I corrected but otherwise it was fine. 

Author Response

Thank you for your review!

We have followed up on all comments made in your attatched pdf-document.

Furthermore, we have followed up on your comments and suggestions as follows:

  1. We have addressed the false negatives in the discussion (4.3), and what we might do to mitigate this issue. We have also pointed out which issues arise with correcting false positives for the other tracked behaviors in the results (3.3).
  2. We have made the results clearer in 3.2 by explaining how many frames of each behavior were labeled. We have also removed a paragraph about the validation percentage of the object detection model in 4.2, since we believe this caused unnecessary confusion and didn't add much to the discussion.
  3. We have written a paragraph (in the Introduction) addressing the background, literature and applications regarding computer vision in this context, where we have clarified the ideas and previous uses of similar methods, along with how our use may differ. We have also discussed this topic in 4.3.

Reviewer 2 Report (Previous Reviewer 2)

Comments and Suggestions for Authors

I am afraid the manuscript is not improved. There are still the same issues as before. Create ML is just a general tool for machine learning, so it cannot be specific on a complex context such as primate behaviours. So it is normal to see the very low accuracy of the model. There are many machine learning algorithms that are created for behaviours on primates, but they are developed based on accelerometer data. I guess the novelty can be the use of video frames, that is probably rare in primates (more common in other animals). If you want to highlight the real novelty you need to reframe the paper. It would be nice to have an overview of other papers using machine learning on video frames to extract behaviours, and which software they used (I reckon R is most common). I think it is a good idea to not include behaviours here, but making this as Communication instead of an article just means that there are not enough details/context. The paper is missing reference to literature.  But I am open to change opinion if the authors can highlight the novelty and put this papers in the right literature context

Comments on the Quality of English Language

NA

Author Response

Thank you for your review!

We have followed up on your comments and suggestions as follows:

  1. We have written a paragraph about previous uses of similar computer vision methods in the introduction, to add some context to the study.
  2. We have reframed the paper to focus on the use of computer vision in the context of using CCTV footage and creating a standardized model that may aid in consistent behavioral coding across studies. 
  3. We have added more details to our methods and explained our results more  clearly to avoid confusion.
  4. We have fixed the mistake of the mismatched references.

Round 2

Reviewer 1 Report (Previous Reviewer 1)

Comments and Suggestions for Authors

This is much clearer. My only edit would be to really flush out your introduction. I, personally, feel as though you've just provided a teeny bit of information on the current use of AI with animal behavior models. You mention some in citations without going into any details about the projects themselves. Just adding a bit more information would be great. 

Otherwise there was one grammatical thing I saw.

Good job! 

Comments on the Quality of English Language

Much better! 

Author Response

Thank you once again for your helpful comments!

We have followed up on the grammatical mistake and have added more information about the cited studies in order to elaborate on the background of similar literature.

We hope this is what you had in mind and that it will be sufficient.

Thank you for your continued interest!

This manuscript is a resubmission of an earlier submission. The following is a list of the peer review reports and author responses from that submission.

Round 1

Reviewer 1 Report

Comments and Suggestions for Authors

I fully believe that this should be two separate papers. The behavioral analysis of visitor presence is one and the integration of the AI model is another. As of right now, the study seems like two separate studies whacked together and is difficult to read and relate to. 

That being said - I think the AI aspect is super interesting and relevant. As we are continuously moving toward automated behavior detection it is always important to evaluate our methodologies like you have. I think, however, you're doing a disservice by having that innovative aspect tacked onto a behavioral study. Especially when you have no validation data attached to it. I think it could be a case study. Ideally, you will have a reliability test between you and the AI model where you both independently score behaviors of the same video and look at how many were scored in the same way. That would be immensely useful for zoo researchers everywhere. 

As of right now, I cannot approve this submission until it is split into two separate papers and rewritten in a way that successfully addresses both topics in a complete manner. Please see attached .pdf for specific comments.

Comments on the Quality of English Language

Be consistent! You switch tenses a few times within the paper so make sure that you are consistent the entire time. Also you shouldn't have "e.g.," or "i.e.," in the middle of a continuing sentence - only when you have a list following them. It is jarring to read when you have those two in the middle of a long sentence. Finally, please read aloud the paper and recognize where some convoluted and run-on sentences exist and fix them. There were quite a few. 

Reviewer 2 Report

Comments and Suggestions for Authors

The concept of the paper is interesting. The paper per se in quite long and there are some parts that are unclear. The main limitation is that the algorithm "object detection" is really far apart from what you get from the algorithm "Manual Ethogram" that is different from what you have in the "Behavioral overview", and all of them are lacking details in the methods so they are not replicable. But especially with "object detection" I see a lack of applicability, too evident the difference with real data. You say that it can autolabel 98.5% of training material, but then it seems that most of it (87%) is just labelled as "out of view/not labelled" so it quite very confusing.  So good idea, but need some clean up and I really struggle to see the applicability of a method with such a bias.

More specific comments:

- need more information on how you collected behavioural data to make fig 5 and 6. 

- R Studio is just an interface of R, need to specify the version of R used

- the methods of machine learning computations cannot be replicated at the moment. You need to add more information for each step. For example, which parameters were used? Which metrics did you get? How can we say that the method was reliable?

- fig 5 and 6 (and others) can you just make a figure where males and females are both in there? Just need % behaviours on the y axis and not x. Also the some figures are not at publication standard 

- fig 7 (and others) need to show a mean and confidence interval for the two groups, now not clear

- fig 10 (and others) This is a creative way of using correlations. Not sure what is the value of correlating between days with the same condition. Is there a reasoning for this? Why not simply doing a comparative test like a Wilcoxon.  

- section 3.2.3 this is not clear. There should be a method associated to this. Not replicable at the moment. Also it seems that the object detection automatic recognition is not really working. There is too much difference with the manual ethogram

- appendix D, something wrong with the output of Wann-Whitney U test, you usually just have U value and p value. Also why a chi square test? 

- why so many figures in the appendix? try to make some panels with figures, merge some figures, etc. now it seems more of a dissertation than a paper.

Comments on the Quality of English Language

na